# Brief Magnetic Field Exposure Stimulates Doxorubicin Uptake into Breast Cancer Cells in Association with TRPC1 Expression: A Precision Oncology Methodology to Enhance Chemotherapeutic Outcome

**DOI:** 10.3390/cancers16223860

**Published:** 2024-11-18

**Authors:** Viresh Krishnan Sukumar, Yee Kit Tai, Ching Wan Chan, Jan Nikolas Iversen, Kwan Yu Wu, Charlene Hui Hua Fong, Joline Si Jing Lim, Alfredo Franco-Obregón

**Affiliations:** 1NUS Centre for Cancer Research, Yong Loo Lin School of Medicine, National University of Singapore, Singapore 117599, Singapore; e0309202@u.nus.edu (V.K.S.); csilsjj@nus.edu.sg (J.S.J.L.); 2BICEPS Lab (Biolonic Currents Electromagnetic Pulsing Systems), National University of Singapore, Singapore 117599, Singapore; nikolas.iversen@u.nus.edu (J.N.I.); lesleywu@nus.edu.sg (K.Y.W.); surcfhh@nus.edu.sg (C.H.H.F.); 3Institute of Health Technology and Innovation (iHealthtech), National University of Singapore, Singapore 117599, Singapore; 4Department of Surgery, Yong Loo Lin School of Medicine, National University of Singapore, Singapore 119228, Singapore; surccw@nus.edu.sg; 5Experimental Therapeutics Programme, Cancer Science Institute, Singapore 117599, Singapore; 6Department of Medicine, Yong Loo Lin School of Medicine, National University Singapore, Singapore 119228, Singapore; 7Department of Haematology-Oncology, National University Cancer Institute, National University Hospital, Singapore 119074, Singapore; 8Department of Physiology, Yong Loo Lin School of Medicine, National University of Singapore, Singapore 117593, Singapore

**Keywords:** personalized medicine, precision medicine, adjunct cancer therapy, DOX contraindications, pulsed electromagnetic fields, anthracycline, adriamycin, targeted therapy, chemoresistance, chemosensitivity

## Abstract

Doxorubicin is a widely used chemotherapeutic agent for breast cancer but is accompanied by significant side effects due to its systemic delivery. The expression of the transient receptor potential canonical 1 (TRPC1) cation channel subunit correlates with breast cancer progression. This study showed that brief magnetic exposure (10 min) increased doxorubicin uptake into breast cancer cells without harming healthy cells. Heightened TRPC1 expression was correlated with more advanced breast cancer grades as well as with greater doxorubicin uptake. Pharmacologically or genetically silencing TRPC1 activity reduced magnetically induced doxorubicin uptake, whereas overexpression of TRPC1 amplified doxorubicin uptake and increased cancer cell death. This study described a localized and non-invasive magnetic therapy paradigm that could potentially improve breast cancer chemotherapeutic efficacy with less systemically delivered doxorubicin. The loading of TRPC1-enriched cell-derived vesicles with doxorubicin upon magnetic exposure underscored the contribution of TRPC1 in the stimulated uptake of doxorubicin in a minimalized model system.

## 1. Introduction

Doxorubicin (DOX) is a mainstay chemotherapeutic agent used for the treatment of a variety of cancers including ovarian, thyroid, bladder, lung and breast cancers as well as for the clinical management of leukemias, lymphomas and sarcomas. In clinical practice, DOX is employed either as a standalone treatment or in combination with other therapies [1]. The efficacy of DOX across a wide range of cancers stems from its broad-spectrum mechanism of action. The anticancer activity of DOX has been attributed to its ability to intercalate between DNA base pairs, effectively disrupting topoisomerase-mediated DNA repair and replication and in the process inhibiting cell proliferation and promoting apoptosis [2]. The intercalation of DOX into mitochondrial DNA disrupts cellular respiration, inhibits mitochondrial biogenesis and produces oxidative stress, which result in ferroptosis and apoptosis [2,3]. In this manner, DOX cytotoxicity is corelated with the mitochondrial density and basal respiratory capacity of a given tissue. DOX exhibits an apparent preference for cancer given the heightened rate of DNA replication associated with the disease, offering greater opportunities for DOX to intercalate into the exposed base pairs [4]. Nonetheless, DOX cytotoxicity is largely agnostic as it ultimately implicates healthy tissues throughout the body, particularly those tissues exhibiting elevated basal levels of mitochondrial respiration [5] such as cardiac [6] and skeletal [7] muscle and central nervous tissues [8]. Consequently, DOX chemotherapy is associated with undesirable side effects with system-wide ramifications [4]. Cardiomyopathy is the principal dose-restricting side effect of DOX therapy that can lead to heart damage and congestive heart failure [9]. DOX chemotherapy also causes skeletal muscle atrophy downstream of mitochondrial dysfunction [10]. The central and peripheral nervous systems are also susceptible to DOX mitotoxicity, leading to cognitive impairment, brain anatomical changes and peripheral neuropathy [11]. Other DOX-associated collateral side effects include myelosuppression, gastrointestinal toxicity and alopecia [4], which have in common tissues of origin exhibiting accelerated rates of cell division [12]. Recent attempts at circumventing systemic DOX toxicity have revolved around the engineering of liposomes to encapsulate free DOX (doxorubicin hydrochloride) for targeted tumor delivery [13]. Although liposomal DOX delivery has been shown to reduce cardiotoxicity, the overall improvements in cancer outcomes have been modest and accompanied with the appearance of new side effects, which places renewed limitations on dose and frequency of administration [13]. The unmet need was therefore a chemotherapeutic strategy that is capable of mitigating DOX-associated side effects without compromising, or even improving, DOX efficacy.

The development of cancer is commonly rooted within cell cycle dysregulation that ultimately translates into tumor initiation, transformation and progression [14]. In accordance, the altered activity of certain TRP (transient receptor potential) channels has been implicated in various forms of cancers [15,16,17,18,19,20]. TRP channels subserve roles in tissue differentiation by triggering downstream enzymatic and transcriptional cascades in response to diverse developmental stimuli [21]. In the context of cancer, the dysregulation of certain TRP channels has been shown to drive oncogenesis due to their maladapted role in executing malignant cell behaviors such as proliferation, migration, epithelial-mesenchymal transition (EMT), invasion and metastasis [18,19,22,23,24]. TRPC1 (transient receptor potential canonical 1) is one of the most ubiquitously expressed of all TRP channel subunits [25,26,27,28]. TRPC1-mediated Ca^2+^ entry typically regulates cell cycle progression in healthy tissues [29,30,31,32,33]. Consequently, the regulation of TRPC1 has been implicated in oncogenic proliferation and progression [34] as well as having been shown to promote tumorigenesis via the PI3K/AKT [23,35,36,37] and ERK1/2 [30] signaling pathways governing cellular proliferation, differentiation and survival [38]. Additionally, it has been shown that TRPC1 modulates TGF-β mediated EMT through store-operated calcium entry (SOCE)-mediated Ca^2+^ entry and calpain activation [34,39]. Elevated TRPC1 expression in cancer cells has also been associated with increased invasiveness and malignancy [34,40]. Notably, elevated TRPC1 expression correlates with heightened susceptibility of aggressive cancers to DOX [34] due to the accelerated growth TRPC1 confers [41]. TRPC1 expression is hence predictive of sensitivity to DOX [22], whereas chemoresistance is associated with downregulation of TRPC1 in breast cancer [22] and ovarian cancer [42]. The TRPC1 channel is hence a viable therapeutic target due to its intimate involvement in key aspects of cancer progression and chemoresistance.

Pulsed electromagnetic fields (PEMFs) have been shown to activate TRPC1-mediated Ca^2+^ entry and downstream catalytic responses in healthy [43] and breast cancer [44] cells. In healthy muscle cells, a brief 10 min exposure to 1.5 millitesla (mT) PEMFs enhanced muscle cell proliferation and differentiation, whereas reducing TRPC1 channel function via pharmacological intervention or genetic silencing precluded magnetic induction [43]. By contrast, in metabolically disrupted breast cancer cells, 3 mT amplitude PEMFs applied for 1 h/day for 3, or more, consecutive days significantly inhibited breast cancer cell proliferation [22,45], whereas healthy cells were unaffected by the same treatment. Importantly, analogous PEMF exposure of breast cancer cells potentiated the anti-cancer effects of DOX *in vitro*, *ex vivo* and *in vivo*, revealing a synergistic relationship between PEMFs, TRPC1 expression and DOX treatment [22].

This study sought to elucidate the potential mechanisms for synergism between PEMF and DOX therapies. The findings from this study (1) validated heightened TRPC1 expression in human breast cancer biopsies, (2) demonstrated enhanced DOX uptake upon PEMF exposure in association with TRPC1 expression in cells and cell-derived vesicles and (3) demonstrated that healthy tissues are less susceptible to PEMF-mediated DOX uptake. The detected synergism between PEMF exposure and DOX chemotherapeutic efficacy may pave the way for the development of new clinical strategies aimed at reducing DOX dosages, especially in cancers characterized by TRPC1 overexpression.

## 2. Materials and Methods

### 2.1. Patient Samples

Voluntary female breast cancer patients who were not pregnant and were over the age of 21 years old were recruited for this study. Healthy and breast tumor tissue samples were post-operatively collected from breast cancer patients as part of a mastectomy or a wider local excision. The patients provided their written informed consent to participate in this study during clinical consultation. This study has been approved by the National Healthcare Group Domain Specific Review Board (2014/01088).

### 2.2. IHC Staining

Breast tissues were maintained in a standard tissue culture incubator for 24 h before overnight fixation with 4% PFA. Subsequently, the tissues were processed using a routine histology protocol involving a graduated ethanol series (50%, 75%, 90% and 100%), clearing using a xylene substitute (Merck, Sigma Aldrich, St. Louis, MO, USA) and embedding in paraffin, whereafter they were sectioned (5 µm) for IHC analysis. After a standard deparaffinization protocol of xylene substitute and ethanol series, antigen retrieval was conducted in Tris-EDTA solution (pH 9) at 60 °C for 30 min. The samples were blocked with Tris-buffered saline (pH 7.4) with Tween-20 supplemented with 5% goat serum and 5% BSA (Merck, Sigma Aldrich). The samples were then stained with the VECTASTAIN^®^ Elite^®^ ABC-HRP kit (Vector Laboratories, Newark, CA, USA) per the manufacturer’s protocol, followed by a Harris Hematoxylin counterstain (Merck, Sigma Aldrich). Primary monoclonal mouse TRPC1 antibody (E-6) (1:50; catalogue no.: sc-133076; Santa Cruz Biotechnology, Dallas, TX, USA) and monoclonal rabbit antigen Kiel 67 (Ki-67) antibody (1:400; catalogue no.: #9129; Cell Signaling Technologies, Danvers, MA, USA) were used. The slides were mounted using VectaMount (Vector Laboratories) and captured under a standard light microscope.

### 2.3. Cell Culture and Pharmacological Reagents

C2C12 mouse skeletal myoblasts were obtained from the American Type Culture Collection (LGC Standards, Teddington, UK) and maintained in DMEM (HyClone, Danaher, Washington, DC, USA) with 10% FBS (Hyclone; Thermo Fisher Scientific, Waltham, MA, USA). Murine 4T1 breast cancer cells were acquired from A/Prof Lina Lim Hsiu Kim’s lab (NUS) and adapted to grow in DMEM with 10% FBS. MCF7 cells were acquired from the American Type Culture Collection and maintained in RPMI 1640 (Gibco, Waltham, MA, USA) supplemented with 10% FBS. TRPC1-GFP overexpressing MCF7 generated previously [22] was maintained in RPMI 1640 containing 500 µg/mL geneticin (Invitrogen, Waltham, MA, USA) and 10% FBS; it is referred to as MCF7-TRPC1 in the manuscript. All cell lines except MCF7-TRPC1 were cultured without antibiotics in a standard tissue culture incubator. The differentiation of myoblasts was induced with a change in medium serum composition from 10% FBS to 2% horse serum (HyClone; Thermo Fisher Scientific) at 24 h after plating at 6000 cells/cm^2^. Doxorubicin hydrochloride (DOX) (Abcam, Boston, MA, USA, ab120629) was reconstituted in DMSO to make a stock concentration of 25 mM and stored at −20 °C. Subsequent dilutions of DOX were made in PBS to keep DMSO concentration below 0.01%.

### 2.4. DOX Standard Curve and Uptake Assay

A standard concentration curve for DOX in RIPA cell lysis buffer was constructed using DOX at 0 nM, 100 nM, 250 nM, 500 nM, 1000 nM and 2500 nM. Absorbance was read at 480/560 nm using a Cytation 5 microplate reader (BioTek, Winooski, VT, USA) and plotted against their respective concentrations. MCF7 and MCF7-TRPC1 breast cancer cells pre-seeded in p10 culture dishes were incubated in 500 nM DOX for 5 min and exposed to 3 mT PEMFs for 10 min. 4T1 breast cancer cells pre-seeded in a p10 culture dish were treated with 50 µM SKF-96365 (Millipore Sigma, Burlington, MA, USA) for 15 min prior to the addition of 500 nM DOX for 5 min. The plates were then exposed to 10 min or 30 min of magnetic exposure at 3 mT in the downward direction. Immediately after exposure, MCF7, MCF7-TRPC1 or 4T1 cells were washed twice in ice-cold PBS, lysed in ice-cold RIPA buffer for 10 min and the collected lysate analyzed at 480/560 nm using Cytation 5 microplate reader (BioTek). Intracellular DOX concentration was normalized to total protein concentration (2 μg/μL) determined by Pierce™ BCA Protein Assay Kits (Thermo Fisher Scientific) following the manufacturer’s protocol.

### 2.5. Real-Time qPCR and TRPC1 Silencing

A quantitative reverse-transcription polymerase chain reaction (RT-qPCR) was carried out using the SYBR green-based detection workflow. Briefly, total RNA was harvested from MCF7, MCF7-TRPC1 and 4T1 cells using an RNeasy kit (Qiagen, Hilden, Germany), and 0.5 µg of RNA was reverse-transcribed to cDNA using iScript cDNA Synthesis kit (Bio-Rad, Hercules, CA, USA). Quantification of gene transcript expression was performed using SSoAdvanced Universal SYBR Green (Bio-Rad) on the CFX Touch Real-Time PCR Detection System (Bio-Rad). Relative transcript expression was determined using the 2^−ΔΔCt^ method, normalized to β-actin transcript levels for cells of human origin or β-2-microglobulin (B2M) transcript levels for cells of murine origin. The qPCR primers used were *hTRPC1*, F: 5′-AAG CTT TTC TTG CTG GCG TG, R: 5′-ATC TGC AGA CTG ACA ACC GT; *hβ-ACTIN*, 5′-AGA AGA TGA CCC AGA TCA TGT TTG A, R: 5′-AGC ACA GCC TGG ATA GCA AC, *mTRPC1*, F: 5′TGG GCC CAC TGC AGA TTT CAA, R: 5′- AAG ATG GCC ACG TGC GCT AAG GAG, *mB2M*, F: 5′- GAT GTC AGA TAT GTC CTT CAG CA, R: 5′: TCA CAT GTC TCG ATC CCA GT.

For TRPC1 silencing in 4T1 cells, two pre-designed dicer-substrate short interfering RNAs (dsiRNA, IDT) were used to knock down the expression of TRPC1. Both dsiRNAs targeted the coding sequence of TRPC1 (NM_011643). Transfection of dsiRNA was performed using Lipofectamine 3000 reagent (Invitrogen) as per the manufacturer’s protocol. TRPC1-silenced cells were validated using qPCR 24 h after dsiRNA transfection using primers against *TPRC1*, as indicated above, relative to cells transfected with scrambled dsiRNA.

### 2.6. Western Analysis

Cell lysates were prepared in ice-cold radioimmunoprecipitation assay (RIPA) buffer containing 150 mM NaCl, 1% Triton X-100, 0.5% sodium deoxycholate, 0.1% SDS and 50 mM Tris (pH 8.0) supplemented with protease (Nacalai Tesque, Kyoto, Japan) and phosphatase inhibitors (PhosphoSTOP, Roche, Basel, Switzerland). Cells were lysed for 20 min and centrifuged for 10 min at 13,500 rpm. The protein concentration of the soluble fractions was determined using a BCA reagent (Thermo Fisher Scientific). An amount of 25–50 µg of total protein was resolved using 10% or 12% denaturing polyacrylamide gel electrophoresis and transferred to PVDF membrane (Immobilon-P PVDF, Millipore Sigma). Proteins on PVDF membranes were blocked using 5% bovine serum albumin in TBST containing 0.1% Tween-20 and incubated with the primary antibody in SuperBlock TBS (Thermo Fisher Scientific) overnight at 4 °C. The primary antibodies used were GFP (1:1000; catalogue no.: 50430-2-AP; Proteintech, Rosemont, IL, USA) and GAPDH (1:10,000; catalogue no.: 60004-1-Ig; Proteintech). The membranes were washed in TBST. Anti-rabbit (31460) or anti-mouse (31430) antibodies conjugated to horseradish peroxidase (HRP) were diluted (1:3000, Thermo Fisher Scientific) in 5% bovine serum albumin in TBST and were incubated with the membranes for 1 h at room temperature. The membranes were incubated in SuperSignal West Pico or West Femto chemiluminescent substrate (Thermo Fisher Scientific), detected with Odyssey Fc imaging system (LI-COR Biotechnology, Lincoln, NE, USA) and analyzed using LI-COR Image Studio version 5.2.5 (LI-COR).

### 2.7. Cell Viability Assay Using CyQuant Cell Proliferation Assay Kit

Cell viability was assessed by measuring DNA content (CyQuant Cell Proliferation Assay kit, Thermo Fisher Scientific) according to the manufacturer’s protocol. Briefly, 4T1 cancer cells or C2C12 myoblasts were seeded in a 96-well format with 8 technical replicates per condition. C2C12 myotubes were used in a 24-well format with 4 technical replicates per condition. The pre-seeded cells were similarly treated with DOX for 5 min prior to magnetic stimulation for 10 min (3 mT) and analyzed 24 h later using CyQuant at 480/520 nm on a Cytation 5 microplate reader (BioTek).

### 2.8. Cell Viability Assay Using MTT Cell Proliferation Kit

Cell viability was also assessed by measuring metabolic activity (MTT Cell Proliferation Kit, Roche) according to the manufacturer’s protocol. Briefly, 4T1 cancer cells or C2C12 myoblasts were seeded in a 96-well format with 8 technical replicates per condition. C2C12 myotubes were used in a 24-well format with 4 technical replicates per condition. The pre-seeded cells were similarly treated with DOX for 5 min prior to magnetic stimulation for 10 min (3 mT) and analyzed 48 h later using MTT labelling agent and solubilization solution at 600 nm on a Cytation 5 microplate reader (BioTek).

### 2.9. Cell-Derived Vesicles Preparation

Cell-derived vesicles (CDVs) were generated as previously described [46]. CDVs are derived from lipid raft domains originating from the muscle cell surface after enzymatic treatment followed by mechanical dissociation. Briefly, C2C12 myoblasts were grown in a 75 cm^2^ flask for 24 h to about 60% confluence. Cells were detached and incubated in 2 mL of RPMI 1640 with 10 µM cytochalasin B (Sigma, St. Gallen, Switzerland) per T75 flask. The flasks were subjected to orbital shaking at 280 rpm for 15 min at 37 °C to promote the formation of CDVs. To aid the detachment of vesicles from cells, the flasks were gently tapped, and the supernatant was collected. It was first centrifuged at 700× *g* for 5 min at 4 °C to remove cells and debris, and then centrifuged again at 10,000× *g* for 30 min at 4 °C to enrich for CDVs. The pelleted CDVs were resuspended in 50 µL PBS with or without 12 µM DOX (final concentration). The CDVs were immediately exposed to PEMFs at 1.5 mT for 10 min. Following magnetic exposure, the CDVs were spun down at 10,000× *g* for 30 min at 4 °C, reconstituted in warm PBS and then provided to pre-plated recipient cancer cells for viability assessment.

### 2.10. PEMF Device

Magnetic fields were generated using a PEMF device as previously described [43]. Briefly, the device produces spatially homogeneous, time-varying magnetic fields, consisting of barrages of 20 × 150 μs on and off pulses for 6 ms repeated at a frequency of 15 Hz. The magnetic flux density rose to a predetermined maximal level within ~50 μs (~17 T/s) when driving field amplitudes to 3 mT. All PEMF-treated samples were compared with time-matched control samples (0 mT).

### 2.11. Statistical Analysis

All statistics were plotted using GraphPad Prism version 10 (San Diego, CA, USA) software. Unless otherwise stated, statistical analyses were performed using one-way analysis of variance (ANOVA) to compare the values between two or more groups followed by Šidák’s multiple comparison post hoc test. For the comparison between two independent samples, Student’s *t*-test was performed.

## 3. Results

### 3.1. TRPC1 Expression in Human Breast Cancer Correlates with Cancer Grade

Breast tumor and adjacent normal tissues were obtained from post-operative specimens following mastectomy or wide local excision. The formalin-fixed, paraffin-embedded tissue samples were processed for immunohistochemistry (IHC) to assess the protein expression of TRPC1 and Ki-67. Normal breast tissue displayed preserved adipose and epithelial architecture, characterized by minimal immunoreactivity for TRPC1 (Figure 1A, normal breast tissue). By contrast, grade 1 and grade 3 breast cancer showed disrupted cytoarchitecture, displaying enlarged and irregular nuclei size (Figure 1A, grade 1 and 3 breast cancer). TRPC1 channel expression was predominantly membrane-bound and cytoplasmic in these pathological tissues. Quantification revealed that TRPC1 expression was highest in grade 3 tumors and lowest in normal tissue, showing a clear trend of increasing TRPC1 expression with advancing cancer grades (Figure 1B). Moreover, TRPC1 expression showed correlation with Ki-67 expression (Figure 1A,B), a marker for cell proliferation [47] and cancer staging [48] in breast cancers. This association suggests a potential role for TRPC1 in promoting cancer cell proliferation and progression.

### 3.2. Time-Dependent Accumulation of Intracellular DOX Following Magnetic Field Exposure

Evidence of functional synergism between TRPC1 expression and DOX chemotherapy has been previously demonstrated [22]. DOX entry into cells has been proposed to occur via its passive diffusion across the membrane despite saturation kinetics being reported [49]. Here we explored the possibility that TRPC1-containing channels are involved in the uptake mechanism of DOX into cancer cells and, moreover, could be stimulated with targeted magnetic exposure. To this end, 4T1 murine breast cancer cells were pretreated with 500 nM DOX for 5 min followed by exposure to PEMFs (3 mT) for either 10 or 30 min. Unexposed 4T1 cells (red bars) exhibited a time-dependent accumulation of intracellular DOX from 10 min to 30 min (Figure 2A). Magnetic field exposure (blue bars) further augmented the intracellular accumulation of DOX two-fold relative to time-matched unstimulated controls. Notably, 30 min of magnetic exposure was capable of accumulating DOX within the cancer cell to approximately 40% of its extracellular levels as calculated in accordance with a standard curve (Figure 2C). Similarly, exposure of MCF7 human breast cancer cells to 3 mT PEMFs for 10 min resulted in a 70% increase in the accumulation of intracellular DOX (Figure 2B). Given that a 10 min PEMF exposure significantly enhanced DOX uptake in both human and mouse cell lines, this exposure paradigm was selected to maximize therapeutic efficacy while minimizing potential uptake into healthy cells.

### 3.3. TRPC1 Contributes to PEMF-Mediated DOX Uptake

A pharmacological strategy was next undertaken to elucidate the potential contribution of TRPC1 in PEMF-induced DOX entry. SKF-96365 is an accepted inhibitor of TRPC channels [50]. 4T1 cells were pre-treated with the TRPC channel inhibitor SKF-96365 (50 µM) for 15 min prior to 5 min incubation with DOX (500 nM) and subsequently exposed to PEMFs (3 mT) for 10 min. Pre-treatment of 4T1 cells with SKF-96365 completely abolished the uptake of DOX in response to PEMF exposure (Figure 3A) while non-SKF-96365-treated cells remained unperturbed in their accumulation of DOX. To further validate the role of TRPC1 in the magnetic induction of cellular DOX entry, transient knockdown of TRPC1 was performed in 4T1 cells using two independent dsiRNAs, achieving approximately 50% reduction in TRPC1 transcript levels (Figure 3B). DOX uptake in TRPC1 knocked-down cells revealed a comparable level of inhibition of DOX uptake (Figure 3C), similar to that observed with pharmacological inhibition. Finally, an established MCF7 cell line stably overexpressing GFP-TRPC1 (MCF7-TRPC1) was employed to confirm the necessity of TPRC1 for DOX uptake. This cell line exhibited elevated TRPC1 expression at both mRNA (Figure 3D) and protein levels (Figure 3E) [22]. Notably, the MCF7-TPRC1 cell line demonstrated enhanced DOX uptake upon magnetic exposure (Figure 3F), with a two-fold increase in PEMF-induced DOX uptake compared to control MCF7 cells. Taken together, these findings demonstrate that TRPC1 is essential for PEMF-induced DOX uptake, highlighting its value as a biomarker with which to determine the potential efficacy of magnetic field therapy for certain cancers.

### 3.4. Synergistic Effects of Magnetic Exposure and DOX Administration on Cancer Cell Viability

Cellular DNA content and metabolic activity were evaluated in 4T1 cells to assess the therapeutic potential of PEMF-induced DOX uptake (Figure 4A,B). The IC_50_ (half maximal inhibitory concentration) for DOX in 4T1 cells at 24 and 48 h were 506.7 nM and 228.4 nM, respectively. Remarkably, a 10 min magnetic exposure (3 mT) in the presence of DOX significantly enhanced the efficacy of DOX, reducing the IC_50_ to 279.9 nM (24 h, CyQuant) and 124.8 nM (48 h, MTT). Notably, the greatest degree of synergism was observed in the lowest concentration range of DOX (50 nM) (Figure 4C,D), where magnetic exposure significantly reduced cell viability by an additional 30% (MTT) and 20% (CyQuant) compared to unstimulated controls receiving the same DOX dose. As the synergy between PEMF and DOX treatments had been previously established in MCF7 and MCF7-TRPC1 cells [22], it was not re-examined in the present study.

The potential for collateral toxicity is of important consideration, particularly as TRPC1 is ubiquitously expressed across tissues [25,26,27,28]. The consequences of the magnetic induction of DOX uptake into a murine muscle cell line (C2C12), before and after myogenic differentiation, was evaluated as an indication of potential collateral cytotoxicity. Employing the same experimental protocol as conducted in cancer cells (Figure 4), no significant difference in viability between stimulated and unstimulated myoblasts was detected (Figure 5A–D). Similarly, no significant difference in viability was detected between stimulated and unstimulated differentiated myotubes (Figure 5E–H). Furthermore, 1.5 mT of magnetic exposure, which coincides with the optimal magnetic strength for myogenic enhancement [43], did not show any enhancement in DOX-mediated cytotoxicity across varying DOX concentrations (Appendix A). Finally, a slight increase in IC_50_ values in myoblasts was apparent with magnetic stimulation (0 mT = 581.5 nM, 1.5 mT = 644.3 nM, 3 mT = 651.6 nM), suggestive of a slight protective effect of magnetic field stimulation against DOX cytotoxicity [43].

### 3.5. Magnetic Exposure Loads TRPC1-Enriched Cell-Derived Vesicles (CDVs) with DOX for Enhanced Cancer Killing

A method was developed for the generation of cell-derived vesicles (CDVs) that are enriched in TRPC1 expression and retain magnetosensitivity [46]. Brief exposure to low energy magnetic fields could load these CDVs with Ca^2+^, corroborating membrane integrity. Moreover, the fusogenic attributes of these CDVs, conferred by the enrichment of the outer leaflet of their phospholipid bilayer membranes with phosphatidylserine, enabled their use as a versatile delivery system. Here, we provide evidence that these CDVs could be loaded with DOX upon magnetic exposure (1.5 mT, 10 min) and that, upon their delivery to breast cancer cells, they were capable of inducing DOX-dependent cytotoxicity (Figure 6). This result would suggest that little more than TRPC1 and magnetic exposure was required for the loading of DOX into a greatly minimized vesicular system (mean diameter of 300 nm) that was largely devoid of cellular reticular structure. Neither magnetic exposure of CDVs alone (−DOX, 1.5 mT; 1st blue) nor incubation of CDVs with DOX alone (+DOX, 0 mT; 2nd red) produced CDVs capable of compromising cancer cell viability. That is, the simultaneity of magnetic and DOX exposures was necessary for effective CDV-mediated DOX delivery. Analogous results were obtained with the human breast cancer cells (MDA-MB231) treated with DOX-PEMF-loaded CDVs (Appendix A). The presence of TRPC1 per se thus appears to be sufficient to support DOX entry upon magnetic exposure.

## 4. Discussion

The etiology of cancer involves a complicated interplay between genetic, environmental and lifestyle factors [51]. At the genetic level, cancer is driven by either the activation of oncogenes and/or the inhibition of tumor suppressors [52,53]. Oncogenes promote cancer when mutated or overexpressed, whereas tumor suppressors inhibit cell growth and division. The activation of oncogenes, or the inhibition of tumor suppressors, can lead to oncogenic transformation and the development of cancer.

Several TRP channel classes have been implicated as oncogenic drivers in a variety of cancers [19,54]. The mammalian TRP channel superfamily consists of 28 cation channel subunits that are subdivided into six subfamilies, based on sequence homology, that include the founding canonical TRP (TRPC), vanilloid TRP (TRPV), melastatin-related TRP (TRPM), ankyrin TRP (TRPA), mucolipin TRP (TRPML) and polycystic TRP (TRPP) subfamilies [55]. Functional TRP channel complexes are generally formed as tetramers of individual TRP channel subunits of familial homomultimeric, or heteromultimeric, stoichiometry [55]. TRP channels commonly receive, integrate and transduce biophysical and chemical stimuli into developmental responses that include cell growth, proliferation and survival [21,56]. Although TRP channels are not inherently oncogenic per se, certain TRP channels have been found to be overexpressed or deregulated in cancer, thereby contributing to oncogenic transformation and tumor progression [18,20,57,58].

Numerous TRP channels have been implicated in the development of certain cancers [19,54] to the point where their expression is being considered prognostic for the cancer in question [15,19]. For instance, the elevated expression of TRPM2 is correlated with breast, lung and prostate cancer cell survival [59], whereas TRPV6 overexpression has been reported in prostate cancer [60] as well as incriminated in breast cancer metastasis [61,62]. Critically, TRPC1 has been shown to contribute to the aberrant Ca^2+^ signaling that confers upon healthy cells oncogenic features such as enhanced migration, invasion, proliferation and survival [34,39,40]. The TRPC subfamily is the most ubiquitously expressed of the TRP superfamilies [26,27,28], with the TRPC1 subunit being the most predominantly expressed member of all [55,63]. The TRPC1 subunit is hypothesized to regulate the activity of the other TRP channel subunits within a channel multimer [64], uniting the unique gating sensitivities of the individual subunits into a single channel complex [65] for the effective integration of diverse biophysical stimuli into a unified developmental response [21]. Accordingly, the deregulated expression of TRPC1 correlates with breast, pancreas and lung cancers [34]. TRPC1 expression may thus serve as prognostic for breast cancer given its strong correlation to tumor progression, metastasis and EMT [34,39,40]. For instance, Figure 1 shows that TRPC1 expression was positively correlated with the grade of breast cancer, indicating correspondence with more aggressive tumor growth and poorer patient outcomes [66], as well as Ki-67 staining, an indicator for poor breast cancer prognosis [67,68]. As Ki-67 is a well-established marker of cellular proliferation, its co-expression with TRPC1 underscores the role of TRPC1 in proliferative signaling pathways. Elevated TRPC1-mediated Ca^2+^ entry also has been shown to stimulate known oncogenic processes including MAPK/ERK [69,70,71], PI3K/AKT [23,35,36] and Wnt/β-cathenin [35] signaling. These proliferative pathways are routinely hijacked and aberrantly commandeered in cancers [72,73,74]. By contrast, loss of TRPC1 channel expression has been shown to attenuate proliferation, migration, invasion and stemness in cancer [22,37]. The cellular mechanisms that upon disruption confer oncogenic properties onto TRPC1 warrant further investigation.

TRPC1 also appears to regulate the activity of the tumor-associated macrophages (TAMs) within the tumor microenvironment (TME). TAMs can assume either pro-tumorigenic or anti-tumorigenic states [75]. Whereas the tumor cancer cell secretome polarizes TAMs towards the pro-tumoral state [76], TRPC1-mediated Ca^2+^ entry into TAM promotes polarization towards the anti-tumoral state [77]. DOX has also been shown to polarize macrophages towards the anti-tumoral state [78]. The ability of PEMFs to activate TRPC1 residing on TAMs may hence be exploited to enhance their loading with DOX to decisively polarize them into the anti-tumor state. Furthermore, localizing the PEMF exposure specifically to the tumor may ultimately allow the lowering of systemic DOX administration (as localized DOX uptake may be enhanced) and will be pursued in future studies.

### 4.1. Implications of TRP Channels in Chemosensitivity

TRP channels have also been implicated in diverse responses to chemotherapy [79]. The inhibition of TRPA1 channel activity by pharmacological means, or by gene silencing, increased cancer sensitivity to carboplatin in *in vitro* models of lung and breast cancer [80]. Similarly, the inhibition of TRPM2 enhanced the sensitivity of breast cancer cells to DOX [81]. Conversely, TRPV2 activation using chemical agonists increased DOX uptake and sensitivity [82]. TRPC3 [83] and TRPC6 [84] have been implicated in DOX-mediated cardiotoxicity, whereas TRPC5 has been associated with P-glycoprotein-induced chemoresistance in cancer cells [85]. TRPC1 expression has also been shown to predict sensitivity to DOX in breast cancer cells [22]. Here, we show that a brief magnetic field exposure enhances DOX uptake in correlation with TRPC1 expression, rendering breast cancer-specific cytotoxicity. Given the fundamental role of TRPC1 in cell proliferation [44], TRPC1 represents a promising prognostic marker for breast cancer. Moreover, the positive correlation between TRPC1 expression and cancer progression provides a therapeutic window of opportunity; TRPC1 expression can be leveraged with magnetic field treatment in combination with DOX administration to achieve greater anti-tumor outcomes. Targeting TRPC1, a key cancer-supporting factor, may confer enhanced selectivity to this therapeutic approach.

Recent efforts to mitigate DOX toxicity have focused on developing next generation DOX strategies, such as DOX encapsulation into liposomes for targeted delivery [13]. Liposomal DOX formulations are designed to exploit the leaky tumor vasculature, thereby increasing drug delivery to the tumor interstitium. Notably, two liposomal DOX formulations, DOXIL and Myocet, have received European regulatory approval, with DOXIL also being granted FDA-approval. These liposomal DOX formulations have been shown to be effective in reducing typical DOX-associated side effects without compromising efficacy. On the other hand, they create a novel set of dose-limiting side effects such as mucosal and cutaneous toxicity [86]. Moreover, the cost of liposomal DOX is ~100 times greater than that of free DOX, restricting its clinical adoption and limiting its potential use [87]. Despite the advances made with liposomal DOX delivery, significant challenges remain to be addressed.

### 4.2. Selective Exploitation of Magnetic Mitohormesis for Anti-Cancer Therapy

A magnetically induced TRPC1-mitochondrial signaling axis was previously demonstrated in normal [43] and breast cancer cells [22] and has potential for clinical exploitation. Magnetic fields invoke a mitohormetic mechanism of action whereby a moderate increase in oxidative stress adaptively strengthens the antioxidant defenses of a cell and promotes its survival, whereas an excessive amount of oxidative stress overwhelms the existing antioxidant defenses of the cell and undermines its survival [88]. The switching point from one effect mode to the other is context dependent. Specifically, the ultimate effect that magnetically induced ROS has on cell fate will depend on the underlying inflammatory status that would then be additive to any supplemental induction of oxidative stress imposed on the cells in question. In this regard, it was shown that moderately strong magnetic field exposure selectively induced cell cycle arrest in breast cancer cells, but not in healthy cells [45], in a manner that was predicted by TRPC1 expression [22]. Implicit to this differential effect are the inherent differences in ROS buffering capacities existing between cancer and healthy cells that provide a therapeutic window of opportunity, wherein cancer cells are more vulnerable to ROS-induced damage while healthy cells remain unharmed due to greater available buffering capacity [89,90].

In muscle cells, brief exposure to low amplitude magnetic fields (1.5 mT, 10 min) supports adaptive cell growth [43], whereas in cancer cells higher amplitude exposure for longer periods (3 mT, 1 h) inhibits malignant cell growth, without affecting healthy cells or tissues [22,45]. This demonstrates the unique ability of this technology to exploit either adaptive (muscle) or damaging (cancer) mitohormetic responses for different objectives by simply changing the amplitude and time of exposure when taken in the context of cellular inflammatory status.

### 4.3. Localized PEMF Exposure of Breast Tumors May Permit Reduction in Systemic DOX Delivery

Extending the magnetic utility of the anti-cancer paradigm described above, the present study revealed that a brief exposure to moderately high-amplitude PEMFs (3 mT, 10 min) significantly further enhanced DOX-mediated cancer cell killing and identified TRPC1 as a crucial mediator of magnetically induced DOX uptake. DOX is typically administered intravenously at a dose of 60 mg/m^2^ [91] and was estimated to achieve a maximum blood concentration (C_max_) of ~5 µg/mL (8.6 µM) [92,93,94]. On the other hand, mathematical modelling has predicted DOX concentrations in plasma and breast adipose tissue to be on the order of 1 µM and 860 nM, respectively [92,93]. Adipose DOX concentration was further calculated to drop by 80% (172 nM) over the next 48 h after administration. Here, we report an *in vitro* DOX IC_50_ of 507 nM for breast cancer cells that was reduced by almost half (280 nM) by magnetic exposure (3 mT for 10 min) over 24 h (Figure 4). Based on these preliminary findings, the standard clinical dose of 60 mg/m^2^ may be theoretically reduced by half, or 30 mg/m^2^, to achieve a plasma C_max_ of 500 nM and adipose accumulation of 430 nM, which would fall within our determined magnetic efficacy window (Figure 4). Accordingly, in a patient-derived xenograft mouse model it was shown that PEMF preconditioning reduced the requirement of DOX requirement for tumor resorption [22], suggestive of such an interaction, albeit not conclusive since the timings for the two interventions did not coincide. Hypothetically, magnetic therapy applied to an inflicted breast could potentially permit a reduction of DOX administered systemically by enhancing its local uptake into the cancer cells within the realms of the magnetic fields. This provocative eventuality remains to be clinically validated in human trials. Nonetheless, given the generalized toxicity of DOX [4], any reduction in its clinical dose could contribute to the reduction of DOX-mediated collateral side effects experienced by patients.

### 4.4. Conventional DOX Chemotherapy Undermines Muscle Resilience to Combat Cancer

The muscle secretome is the body’s first line of defense against cancer [95]. The muscle secretome is a downstream response limb of mitochondria respiration [96,97,98] and, as such, will be compromised by DOX-dependent muscular mitochondrial disruption [5,10]. Paradoxically, components of the muscle secretome have been shown to be protective against DOX-induced cardiotoxicity [99], whose protective action would be undermined by DOX therapy. In principle, any therapeutic strategy that would permit the lowering of systemic DOX dosage should yield far-ranging and mutually reinforcing clinical benefits.

DOX intercalates within mtDNA [3] to trigger mitochondrial dysfunction and oxidative stress [100]. The formation of DOX-Fe complexes also generates semiquinone radicals that further contribute to the production of mtROS [3,101]. These DOX-Fe complexes also deplete iron that compromises mitochondria efficiency [102], exacerbating the accumulation of mtROS that further damages mitochondria and results in cell death via ferroptosis. The constitutive uptake of DOX by collateral tissues and the detrimental effect it has over organismal resilience (Figure 5) underscores the clinical need for the development of therapeutic measures to reduce the level of systemic DOX administration. DOX-associated collateral damage preferentially affects organs enriched with mitochondria, such as the heart, skeletal muscle and the central nervous system [6,7,8]. Ultimately, DOX chemotherapy creates a degenerative cycle whereby DOX-mediated muscle atrophy weakens anti-cancer attributes of the muscle secretome, increasing the need for DOX to manage cancer, which further dampens the muscle secretome response.

### 4.5. Combining Muscle and Breast Tumor Anti-Cancer Magnetic Therapies

An adjuvant magnetic therapy has recently been described that targets muscle and fortifies its anti-cancer secretome response [103]. Muscle-targeted magnetic therapy may hence improve systemic resilience against DOX chemotherapy, whereas breast tumor-targeted magnetic therapy may confer vulnerability to lower levels of systemic DOX (see *Selective Exploitation of Magnetic Mitohormesis for Anti-Cancer Therapy*). The deployment of a combinatorial magnetic therapy is hence on the horizon that could exploit the unique anti-cancer magnetic responses of muscle and breast cancer for enhanced chemotherapeutic outcomes. Importantly, muscle cells appear to be less sensitive than breast cancer cells to DOX-mediated cytotoxicity when administered in conjunction with magnetic exposure (Figure 5), possibly reflecting inherently lower expression levels of TRPC1 in healthy muscle. The pairing of muscle (systemic action) and cancer (local action; see *Localized PEMF Exposure of Breasts Tumors May Permit Reduction in Systemic DOX Delivery*) magnetic therapies may allow for a lowering of systemic DOX dosage while maintaining DOX uptake and efficacy at the tumor. Such a combinatorial approach would help spare muscle loss to more fully harness the anti-cancer potential of the muscle secretome and assist in reversing the previously described vicious degenerative cycle. This magnetic platform technology may thus pave the way for a multi-level therapeutic strategy to improve targeted cancer killing, reduce systemic chemotherapy dose and potentially minimize harm to healthy tissues.

### 4.6. Evidence for TRPC1-Mediated DOX Uptake

A protocol for the creation of cell-derived vesicles (CDVs) was developed that produces CDVs that are selectively enriched for TRPC1 and TRPA1 [46]. These CDVs could be loaded with Ca^2+^ upon exposure to PEMFs, satisfying the reported requirement for TRPC1 in magnetoreception. Most notably, these CDVs could restore magnetically induced respiratory capacity to TRPC1 knockdown cells. The loading of DOX into CDVs of a mean diameter 300 nm implies a close relationship between TRPC1 and DOX uptake.

It was previously shown that the fluorescent cationic dye FM1-43 (~12 A° diameter, 611 Da) was capable of being taken up into cells in association with TRPC1 that correlated with cell proliferation [44]. TRPA1 also has been shown to support the entry of FM1-43 into cells [104]. The pore of TRPA1 has an estimated diameter of ~11 A° and has been shown to be dynamically regulated to accommodate large fluorescent molecules, such as FM1-43, upon biophysical activation [105]. TRPA1 has been shown to coimmunoprecipitate with TRPC1 [106], suggesting that TRPC1 and TRPA1 can heteromultimerise to form a functional channel complex [107] of potentially regulatable permeability to larger macromolecules. One possibility, therefore, is that TRPC1/TRPA1 heteromultimers are capable of accommodating DOX entry into cells upon magnetic stimulation. Accordingly, here we showed that these same CDVs could be loaded with DOX upon magnetic exposure that was then capable of transmitting DOX-mediated cytotoxicity to breast cancer cells (Figure 6). The notion that DOX is entering via TRPC1 is provocative but remains to be specifically shown.

### 4.7. Breast Cancer Specificity of Magnetically Stimulated DOX Uptake

In healthy tissues, TRPC1 channel expression is developmentally regulated, being most heightened during the proliferative phase and downregulated after terminal differentiation [43,44]. Breast cancer cells exist in a state of sustained stemness and hence retain aberrantly elevated TRPC1 expression throughout most of the course of the disease [22]. Breast cancer is hence developmentally poised for preferential vulnerability to this magnetic therapeutic strategy that targets TRPC1 expression. Healthy tissues, on the other hand, predominantly restrict the expression of TRPC1 to the early phase of cell proliferation, when it is transient and short lived. Accordingly, Figure 5 and Appendix A show that skeletal muscle, our largest collateral tissue with demonstrated vulnerability to DOX [7], is not implicated in the enhanced vulnerability to DOX provoked by this form of magnetic field exposure. In this respect, our magnetic therapeutic strategy potentially exhibits unprecedented selectivity towards cancers characterized by elevated TRPC1 expression.

## 5. Conclusions

TRPC1 was previously shown to be activated by magnetic field exposure. TRPC1 expression also positively correlates with the proliferative capacity of human breast cancer cells. The present study revealed that TRPC1 expression correlated with the ability of brief magnetic exposure to enhance the uptake of the chemotherapeutic agent, doxorubicin (DOX), into cancer cells. Most notably, the IC_50_ of DOX was reduced by concomitant magnetic field exposure of cancer cells, whereas the IC_50_ of skeletal muscle cells to analogous DOX-PEMF treatment was slightly increased, demonstrating therapeutic specificity. Cell-derived vesicles enriched in TRPC1 were capable of being loaded with DOX upon magnetic exposure, suggesting that TRPC1 per se is sufficient to support DOX entry upon magnetic exposure. Deliberately designed magnetic field paradigms may hence serve as effective adjuvant therapies to systemic DOX chemotherapy by providing a non-invasive and targetable co-intervention. This synergistic therapeutic strategy may, in turn, enable the lowering of clinically administered DOX dosages with the benefit of minimizing DOX-associated side effects. Future clinical trials are warranted to substantiate and further elaborate upon these possibilities.

## Figures and Tables

**Figure 1 cancers-16-03860-f001:**
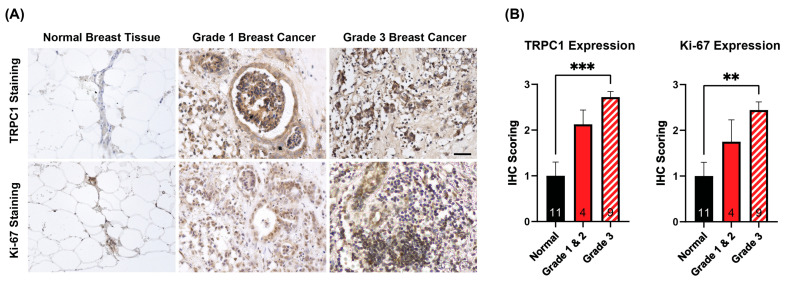
Immunohistochemistry (IHC) staining of breast tumor and neighboring normal breast tissue. (**A**) Representative IHC images showing TRPC1 and Ki-67 staining (brown) in breast tumor (Grade 1 and 3) and neighboring normal breast tissue. Tissue sections were counterstained with hematoxylin (blue). The scale bar is 50 μm. (**B**) Semi-quantitative IHC analysis of TRPC1 and Ki-67 staining intensity. Staining intensity was scored as: 0 (no staining), 1 (weak), 2 (moderate) or 3 (strong staining). Data represent mean ± standard error of the mean (SEM). The number of independent samples is reflected within each bar, and the samples were analyzed using one-way ANOVA, followed by Šidák’s multiple comparison post hoc test. Statistical significance is indicated by **, *p* ≤ 0.01 and ***, *p* ≤ 0.001.

**Figure 2 cancers-16-03860-f002:**
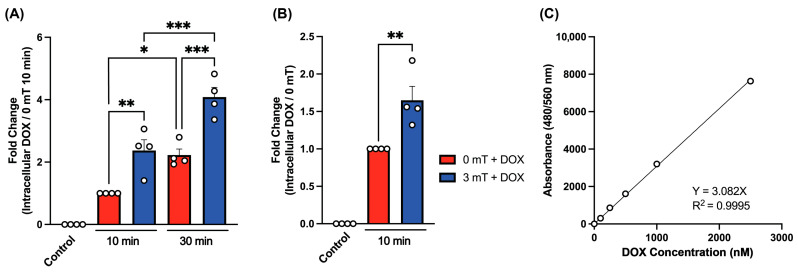
Magnetic field exposure increases intracellular DOX concentration in a time-dependent manner. (**A**) Intracellular DOX concentration (expressed as fold change) of 4T1 murine breast cancer cell line incubated in 500 nM DOX for 5 min prior to magnetic exposure for 10 or 30 min as indicated. Control cells were not treated with DOX or magnetic fields. (**B**) Intracellular DOX concentration (expressed as fold change) of MCF7 human breast cancer cell line incubated in 500 nM DOX for 5 min prior to magnetic exposure for 10 min as indicated. Control cells were not treated with DOX or magnetic fields. The intracellular DOX concentrations were calculated from a standard curve. (**C**) Intracellular DOX concentration generated by measuring the intrinsic autofluorescence of DOX at 480/560 nm absorbance from serial dilutions in lysis buffer. Data represent mean ± standard error of the mean (SEM) (n = 4, with 3 technical replicates) and were analyzed using one-way ANOVA, followed by Šidák’s multiple comparison post hoc test. Statistical significance is indicated by *, *p* ≤ 0.05; **, *p* ≤ 0.01 and ***, *p* ≤ 0.001.

**Figure 3 cancers-16-03860-f003:**
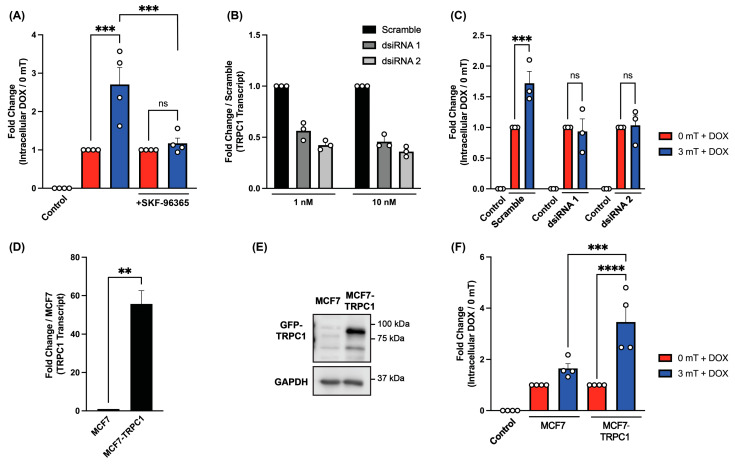
Magnetic field-induced DOX uptake correlates with TRPC1 expression. (**A**) Bar chart showing the fold change in intracellular DOX concentration of 4T1 cells pre-treated with 50 µM SKF-96365 for 15 min and 500 nM DOX for 5 min prior to 10 min magnetic exposure (n = 4). (**B**) Bar chart showing fold changes in TRPC1 transcript levels as detected by qPCR in 4T1 cells transfected with scrambled or TRPC1 dsiRNA after 24 h. Data represent mean ± standard deviation, (n = 3 technical replicates). (**C**) Bar chart showing fold change in intracellular DOX concentration in 4T1 cells transfected with scramble or TRPC1 dsiRNA. Cells were pre-treated with 500 nM DOX for 5 min prior to 10 min magnetic exposure (n = 3). (**D**) Bar chart showing fold change in TRPC1 transcript level detected by qPCR in MCF7 and MCF7 stable cell line overexpressing TRPC1 (MCF7-TRPC1) cells (n = 3). (**E**) Western blot of GFP-TRPC1 in MCF7 and MCF7-TRPC1 cells (n = 4). The uncropped blots are shown in Appendix A. (**F**) Bar chart showing the fold change in intracellular DOX concentration of MCF7 and MCF7-TRPC1 cells pre-treated with 500 nM DOX for 5 min prior to 10 min of magnetic exposure (n = 4). Unless otherwise stated, data represent mean ± standard error of the mean (SEM). Statistical analysis was performed using one-way ANOVA, followed by Šidák’s multiple comparison post hoc test. Significance is indicated by ns (not significant); **, *p* ≤ 0.01; ***, *p* ≤ 0.001; and ****, *p* ≤ 0.0001.

**Figure 4 cancers-16-03860-f004:**
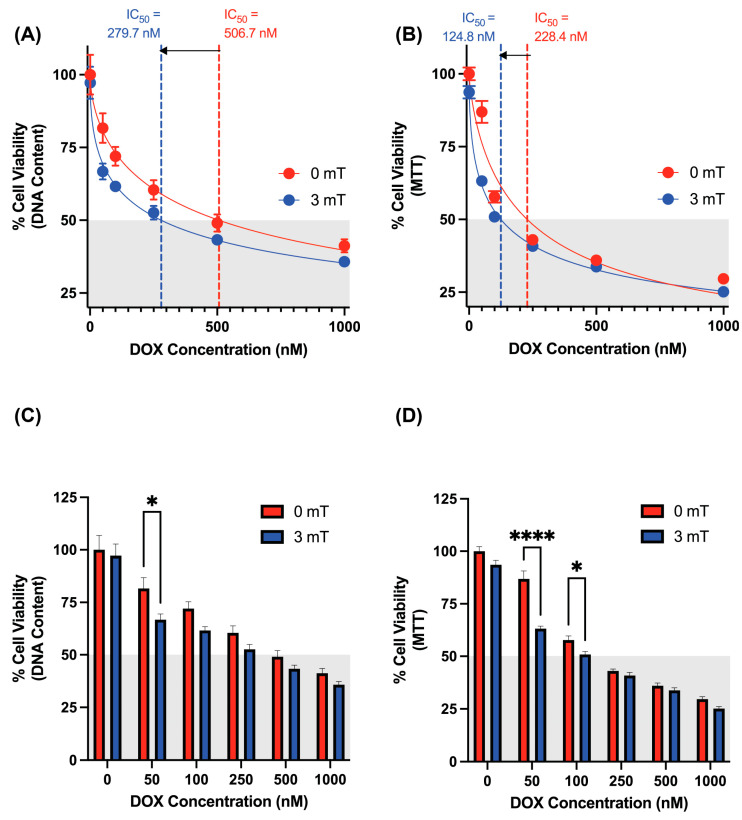
Magnetic exposure enhances DOX-mediated cytotoxicity in 4T1 cells. (**A**) Dose–response curve of 4T1 cells treated with increasing concentrations of DOX for 24 h, with IC_50_ values as extrapolated from the red (unexposed, IC_50_ = 506.7 nM) or blue (magnetically exposed, IC_50_ = 279.7 nM) symbols as generated using cellular DNA content. (**B**) Dose–response curve of 4T1 cells treated with increasing concentrations of DOX for 48 h. IC_50_ values as extrapolated from the red (unexposed) or blue (magnetically exposed) symbols as generated measuring mitochondrial respiration (MTT). (**C**) Bar chart corresponding to panel (**A**) showing % cell viability of 4T1 cells 24 h after treatment with magnetic fields and DOX. (**D**) Bar chart corresponding to panel (**B**) showing % cell viability of 4T1 cells 48 h after treatment with magnetic fields and DOX. Panels (**A**,**C**) were generated using a CyQuant DNA content assay kit as described in Section 2. Panels (**B**,**D**) were generated using an MTT assay kit as described in Section 2. In all cases, DOX was added 5 min prior to magnetic exposure for 10 min. Data represent mean ± standard error of the mean (SEM) (n = 5, with 8 technical replicates for CyQuant; n = 2, 8 technical replicates for MTT). Statistical analysis was performed using 2-way ANOVA, followed by Šidák’s multiple comparison post hoc test. Significance is indicated by *, *p* ≤ 0.05 and ****, *p* ≤ 0.0001.

**Figure 5 cancers-16-03860-f005:**
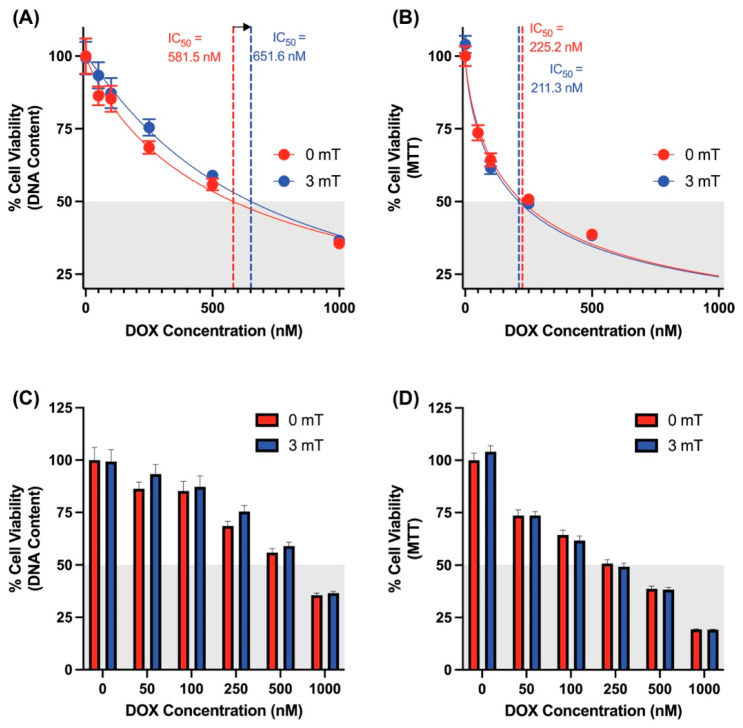
Muscle cells are not susceptible to PEMF-enhanced, DOX-mediated cytotoxicity. (**A**) Dose–response curve of C2C12 myoblasts treated with increasing concentrations of DOX for 24 h, with IC_50_ values as extrapolated from the red (unexposed, IC_50_ = 581.5 nM) or blue (magnetically exposed, IC_50_ = 651.6 nM) symbols as generated using cellular DNA content. (**B**) Dose–response curve of C2C12 myoblast cells treated with increasing concentrations of DOX for 48 h. IC_50_ values as extrapolated from the red (unexposed), or blue (magnetically exposed) symbols as generated measuring mitochondrial respiration (MTT). (**C**) Bar chart corresponding to panel **A** showing % cell viability of C2C12 myoblasts treated with increasing concentrations of DOX for 24 h after treatment with magnetic fields and DOX. (**D**) Bar chart corresponding to panel (**B**) showing % cell viability of C2C12 myoblasts treated with increasing concentrations of DOX for 48 h after treatment with magnetic fields and DOX. (**E**) Dose–response curve of C2C12 myotube cells treated with increasing concentrations of DOX for 24 h. (**F**) Dose–response curve of C2C12 myotube cells treated with increasing concentrations of DOX for 48 h. (**G**) Bar chart corresponding to panel (**E**) showing % cell viability of C2C12 myotubes treated with increasing concentrations of DOX for 24 h after treatment with magnetic fields and DOX. (**H**) Bar chart corresponding to panel **F** showing % cell viability of C2C12 myotubes treated with increasing concentrations of DOX for 48 h after treatment with magnetic fields and DOX. Panels (**A**,**C**,**E**,**G**) were generated using a CyQuant DNA content assay kit as described in Section 2. Panels (**B**,**D**,**F**,**H**) were generated using an MTT assay kit as described in Section 2. In all cases, DOX was added 5 min prior to magnetic exposure for 10 min. Data represent mean ± standard error of the mean (SEM) (n = 4, with 8 technical replicates for myoblasts, 4 technical replicates for myotubes). Statistical analysis was performed using two-way ANOVA, followed by Šidák’s multiple comparison post hoc test. A similar data set generated with the muscle specific exposure paradigm (1.5 mT for 10 min) is provided in Appendix A.

**Figure 6 cancers-16-03860-f006:**
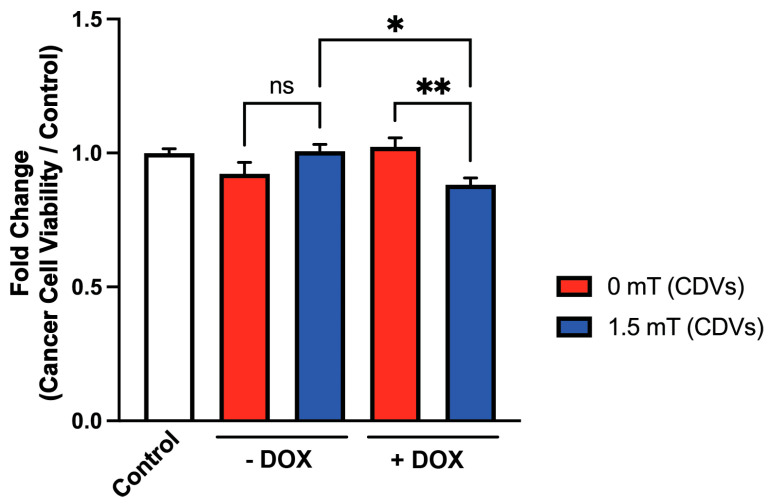
Magnetic exposure loads cell-derived vesicles (CDVs) with DOX that can be then transferred to breast cancer cells for targeted killing. Fold change in the viability of 4T1 murine breast cancer cells after treatment with CDVs that were either magnetically exposed (1.5 mT), or not (0 mT) for 10 min, while in the absence or presence of 12 µM DOX. Each repetition of a condition was provisioned with the quantity of CDVs generated from approximately 1.5 × 10^6^ wild type C2C12 myoblasts. An optimization of the concentration of CDVs to observe the greatest cell response was not attempted here; the only objective was to detect differential cell responses. Cell viability was ascertained by quantifying cellular DNA using CyQuant as described in Section 2. Data represent mean ± standard error of the mean (SEM) (n = 5, with 6 technical replicates). Statistical analysis was performed using one-way ANOVA, followed by Šidák’s multiple comparison post hoc test. Significance is indicated by ns (not significant); *, *p* ≤ 0.05; **, *p* ≤ 0.01. A similar data set generated on MCF-7 breast cancer cells as recipients for the CDVs is provided in Appendix A.

## Data Availability

All data supporting the results are presented in the manuscript. Any other inquiries can be directed to the corresponding authors via email.

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
