# Peer review of "Brief Magnetic Field Exposure Stimulates Doxorubicin Uptake into Breast Cancer Cells in Association with TRPC1 Expression: A Precision Oncology Methodology to Enhance Chemotherapeutic Outcome"

_cancers, 2024, doi:10.3390/cancers16223860_

Round 1
Reviewer 1 Report
Comments and Suggestions for Authors
This manuscript presents results from in vitro experiments to determine how TRPC1 expression in combination with pulsed electromagnetic field (PEMF) exposure influences uptake of doxorubicin (DOX) in breast cancer cells relative to muscle cells. This work builds upon prior experiments from this group to explore therapeutic effects of PEMF in breast cancer cells and xenograft tumors. The effects of short-term exposure to PEMF (10-30min) following DOX treatment were determined on cell viability using two assays (DNA content and MTT assay). The article provides extensive background and citations that relate to the topics presented. Results are clearly portrayed in the figures and are adequately discussed in the text. Major conclusions are supported by the results provided in the manuscript. Review of the manuscript raised the following points and comments for the authors to consider.
Minor comments:
-Figure 2B shows fold-change in DOX, not concentration; please update the caption accordingly.
-Figure 4C,D: the statistical analyses for this data should not use both 2-way ANOVA (with post-test) and unpaired t-test.
Reviewer 2 Report
Comments and Suggestions for Authors
Sukumar et al studied the effect of PMEF on DOX uptake in breast cancer cells. They enhanced DOX uptake with PEMF and this uptake is further increased by increased expression of TRPC1 leading to efficient killing of cancer cells. Furthermore, genetic or TRPC1 antagonist reduced DOX uptake. These results suggests that TRPC1 mediated PEMF could be developed as potential therapy for breast and other cancer where DOX is generally used as a chemotherapeutic agent.
The experiments shown does not support the conclusion exclusively based on design and execution. The following concerns still remained in the study.
1. Line 241, CDVs should be expanded when first described.
2. Line 285-286, no stage 3 tissue is shown in figure1A with anti-TRPC1 but the quantitation are done in figure 1B. They should show them or It should be removed from the figure 1B .
3. Figure 2 A and B, Y-axis is differently labelled. Why 10 minutes or more than 30 minutes mT exposure should be shown to visualize the range of DOX uptake.
4. In line 325, Fig 3C legend, what cells are these?
5. As in Figure 2, the maximum DOX uptake was observed after 30 min exposure of 3m T, then why 10 min exposure were chosen for experiments represented in Figure 3. These needs explanation in the results.
6. There are numerous mistakes in the manuscript, such as some where it is SKF-96365, somewhere SKF96365 (line 336), CDVs (line 419) etc.
7. Line 355, the heading is not appropriate. In these experiments, only DOX concentrations were changed, but the Mag exposure was constant as 3M T for 10 minutes.
8. Figure 4, why these experiments are done only 4T1 mouse cells but not done in MCF-7 cells when all materials are available in hand? They should show in the effect of DOX conc and also exposure time variations in MCF-7 cells.
9. Line 103-107, it appears that TRPC1 upregulation increases cancer growth. Then, how it helps to kill cancer cells with DOX and PMEF? This needs explanation in the discussion.
10. Half of the experiments were done in mice cells and half were done in human MCF-7 cells. But the conclusion is accumulative.
11. Line 140-142, whether consent was taken from individual with normal breast tissue?
12. Line 152, In IHC, what is Ki-67 stands for? It should be mentioned earlier before presenting in the figure.
13. What are the catalogue number of respective secondary antibodies or whether primary antibodies are conjugated? IHC methods should be described in more detail.
14. Line 197, what is mB2M?
15. Line 214, 5% is typed twice
16. Supplementary materials are never referred within the text
Reviewer 3 Report
Comments and Suggestions for Authors
In this study, the authors demonstrated that magnetic exposure could selectively enhance doxorubicin (DOX) uptake in breast cancer cells, particularly those expressing high levels of TRPC1, without affecting healthy cells. They found that TRPC1 expression correlated with breast cancer progression and facilitated DOX uptake, with magnetic exposure further amplifying this effect. By pharmacologically or genetically silencing TRPC1, they confirmed its role in magnetically-induced DOX uptake, while TRPC1 overexpression increased both DOX uptake and cancer cell death. This localized, non-invasive magnetic therapy approach suggests a promising avenue for improving DOX efficacy in breast cancer treatment with reduced systemic toxicity. I have few questions and suggestions.
1) This introduction provides a thorough overview of doxorubicin (DOX) and its mechanisms of action, Here are some refinements for clarity and flow
You might begin by emphasizing why DOX is a mainstay treatment, perhaps mentioning its efficacy across diverse cancer types due to its broad-spectrum action.
The paragraph on liposomal DOX delivery could be expanded by mentioning specific outcomes or clinical trials if applicable, showing how encapsulation improves the therapeutic index of DOX while noting limitations.
2) For the IHC results, explicitly state that TRPC1 expression increases with cancer progression, with minimal expression in normal tissues and highest levels in stage 3 tumors. Also, explain how Ki67 co-expression underscores TRPC1's role in proliferation.
3) What specific mechanisms might underlie the observed increase in DOX uptake in TRPC1-expressing cells under magnetic field exposure? Could this be related to conformational changes in TRPC1 channels or enhanced channel activity in response to electromagnetic fields?
4) Given that magnetic field exposure seems to increase DOX uptake, how feasible would it be to implement this approach clinically? What challenges might arise with the application of pulsed electromagnetic fields in patients, especially concerning field intensity, duration, and targeting specificity?
5) If TRPC1 channels facilitate increased DOX uptake and enhance its cytotoxicity in cancer cells, could TRPC1 be directly targeted or modulated to increase chemotherapy efficacy? Would there be potential risks, given TRPC1’s expression in normal tissues?
6) Increased intracellular accumulation of DOX via TRPC1 and magnetic field exposure could raise safety concerns. How might this approach impact surrounding normal tissues that may also express TRPC1? Are there protective measures that could ensure selective targeting of cancer cells?
7) What limitations might arise in scaling magnetic field therapy for cancers in various anatomical locations, especially those not easily accessible for targeted magnetic exposure? How might different tumor locations impact the effectiveness and design of such treatments?
8) Could prolonged exposure to magnetic fields or increased DOX uptake through TRPC1 channels lead to adaptive resistance in cancer cells? If so, what secondary strategies could be considered to maintain therapeutic efficacy?
Round 2
Reviewer 3 Report
Comments and Suggestions for Authors
The authors successfully addressed all the comments, the manuscript is suitable for publication